# Neuromuscular electrical stimulation for physical function maintenance during hematopoietic stem cell transplantation: Study protocol

**Lindsey J. Anderson**[1,2]*, **Lauren Paulsen**[1], **Gary Miranda**[1], **Karen L. Syrjala**[3,4], **Solomon A. Graf**[3,5,6], **Thomas R. Chauncey**[3,5,6], **Jose M. Garcia**[1,2]

1 Geriatric Research, Education and Clinical Center, Veterans Affairs Puget Sound Health Care System, Seattle, Washington, United States of America, 2 University of Washington Department of Medicine, Division of Gerontology and Geriatric Medicine, Seattle, Washington, United States of America, 3 Clinical Research Division, Fred Hutchinson Cancer Research Center, Seattle, Washington, United States of America, 4 Psychiatry and Behavioral Sciences, University of Washington School of Medicine, Seattle, Washington, Unites States of America, 5 Hospital and Specialty Medicine, Veterans Affairs Puget Sound Health Care System, Seattle, Washington, United States of America, 6 University of Washington Department of Medicine, Division of Medical Oncology, Seattle, Washington, Unites States of America

* lindsey.anderson5@va.gov

**Data Availability Statement:** No datasets were generated or analysed during the current study.

## Abstract

Hematopoietic stem cell transplantation is a common life-saving treatment for hematologic malignancies, though can lead to long-term functional impairment, fatigue, muscle atrophy, with decreased quality of life. Although traditional exercise has helped reduce these effects, it is inconsistently recommended and infrequently maintained, and most patients remain sedentary during and after treatment. There is need for alternative rehabilitation strategies, like neuromuscular electrical stimulation, that may be more amenable to the capabilities of hematopoietic stem cell transplant recipients. Patients receiving autologous HCT are being enroled in a randomized controlled trial with 1:1 (neuromuscular electrical stimulation:sham) design stratified by diagnosis and sex. Physical function, body composition, quality of life, and fatigue are assessed prior to hematopoietic stem cell transplant (prior to initiating preparatory treatment) and 24±5 days post hematopoietic stem cell transplant (Follow-up 1); physical function and quality of life are also assessed 6-months post hematopoietic stem cell transplant (Follow-up 2). The primary outcome is between-group difference in the 6-minute walk test change scores (Follow-up 1—Pre-transplant; final enrolment goal N = 23/group). We hypothesize that 1) neuromuscular electrical stimulation will attenuate hematopoietic stem cell transplant-induced adverse effects on physical function, muscle mass, quality of life, and fatigue compared to sham at Follow-up 1, and 2) Pre-transplant physical function will significantly predict fatigue and quality of life at Follow-up 2. We will also describe feasibility and acceptability of neuromuscular electrical stimulation during hematopoietic stem cell transplant. This proposal will improve rehabilitative patient care and quality of life by determining efficacy and feasibility of a currently underutilized therapeutic strategy aimed at maintaining daily function and reducing the

**Funding:** This study was funded by a grant from the U.S. Dept. of VA (grant number RX003245; PI: Anderson). JMG receives research support from the VA (BX002807), the Congressionally Directed Medical Research Program (PC170059), and the National Institutes of Health (NIH; R01CA239208, R01AG061558). These funders had no role in study design, data collection and analysis, decision to publish, or preparation of the manuscript.

**Competing interests:** The authors have declared that no competing interests exist.

impact of a potent and widely used cancer treatment. This trial is registered with clinical-trials.gov (NCT04364256).

## Introduction

Over 60,000 hematopoietic stem cell transplants (HCT) are performed annually worldwide for hematologic malignancies [1]. This complex process involves high doses of systemic anti-cancer treatment and can lead to both short and long-term deconditioning, fatigue, sarcopenia, and poor overall quality of life (QOL) [2]. Effective mitigation of these issues remains an important area of unmet need in HCT care. Low muscle mass (sarcopenia) is present in up to half of patients prior to HCT, and those with sarcopenia display worse handgrip strength (HGS), quadricep strength, 6-minute walk test (6MWT) distance, and patient-reported physical function and vitality [3]. Functional impairment and sarcopenia are especially problematic because high-dose chemotherapy before HCT is often associated with impaired nutritional status, reduced QOL, and disuse atrophy (inactivity-related muscle wasting) during/after HCT [4,5]. Sarcopenia, functional impairment, and reduced lower body strength prior to HCT have also been independently correlated with prolonged HCT-related hospitalization and/or poorer overall and disease-free survival [6–9].

A recent meta-analysis of randomized-controlled trials reported a significant benefit of traditional resistance/aerobic exercise on muscle strength and patient-reported fatigue and QOL when initiated prior to HCT [10]. Although, most patients remain sedentary through and after the HCT process [4,5] which is largely attributed to inconsistent exercise recommendation practices and lack of adherence to recommended exercise [11]. For example, in a cohort of 201 HCT patients, only 2% were referred for physical therapy during in-patient post-HCT recovery [12]. In addition, implementation of supervised exercise, like traditional resistance exercise, can be especially problematic for HCT patients who are immunocompromised and experience high levels of fatigue. As a result, there is a need for investigating alternative rehabilitation strategies aimed at maintaining physical function which may be more amenable to the needs and capabilities of HCT recipients during the peri-transplant period.

Neuromuscular electrical stimulation (NMES) is a U.S. Food and Drug Administration-approved therapeutic modality for many indications including prevention/attenuation of disuse atrophy. It induces involuntary contraction via cutaneous electrodes placed over target muscle(s) and has benefited physical function in patient populations with similar functional impairments as HCT recipients [13,14]. Stair climb power and 5-times-sit-to-stand time improved in elderly patients when quadriceps NMES was initiated immediately after hip replacement compared to usual care [15]. In chronic obstructive pulmonary disease or congestive heart failure, lower body NMES improved 6MWT, quadriceps strength, quadriceps mass, and/or aerobic capacity compared to sham [16–20]; patient-reported fatigue was also improved in chronic obstructive pulmonary disease compared to sham [20,21]. Patients undergoing HCT often experience similar disuse-associated rapid muscle atrophy and functional decline and moderate cardiopulmonary dysfunction as these patient cohorts and may therefore experience a similar benefit from NMES.

In the cancer setting, NMES has only been tested in small samples with mixed results. A pilot study testing quadriceps NMES in non-small cell lung cancer patients not undergoing anti-cancer treatment reported a trend for improved quadriceps strength compared to usual care [22]. In a small follow-up study, quadriceps NMES in non-small cell lung cancer patients undergoing chemotherapy did not induce group differences for change in quadriceps strength

or mass [23]. However, patient-reported fatigue measured by the Multidimensional Fatigue Inventory (MFI)-20, improved from NMES in that cohort [23]. In another study including patients with advanced solid tumors undergoing anti-cancer therapy, whole-body NMES improved 6MWT to a greater extent than usual care, which was only clinically meaningful for the NMES group [24]. There was also a trend for improved fatigue with NMES measured by the Functional Assessment of Chronic Illness Therapy-Fatigue (FACIT-F) [24].

The use of NMES to combat functional decline, disuse atrophy, and worsened fatigue may be particularly useful during HCT where these symptoms are features of the disease, its initial treatment, and the HCT process. A single-arm feasibility study of quadriceps and biceps NMES was conducted in hematologic cancer patients undergoing intensive chemotherapy with or without HCT [25]. The changes for 6MWT (-24m) and MFI-20 fatigue (-1 point) were not clinically meaningful [26,27], indicating a potential maintenance effect, and NMES was well-tolerated with no major bleeding events, cardiac arrhythmia, or rhabdomyolysis measured by creatine kinase (CK) with ~65–70% of patients achieving the adherence standard (66% of the intended training time) [25]. While NMES may be safe and well-tolerated during HCT, its potential to benefit physical function and fatigue during HCT has yet to be tested in a randomized controlled setting. It is also unknown whether stimulating multiple lower body muscle groups and reducing prescription to three days/week to allow muscle recovery between sessions would induce a clinically meaningful impact on physical function during HCT.

The overall goals of this randomized, sham-controlled study are to assess the 1) efficacy of NMES vs sham on HCT-induced functional decline, muscle atrophy, and worsening of patient-reported fatigue and QOL, 2) association between Pre-HCT physical function and prolonged recovery of patient-reported fatigue and QOL, and 3) feasibility and acceptability of NMES during the peri-transplant period. We hypothesize that 1) NMES will attenuate the acute HCT-induced negative impact on physical function, body composition, QOL, and fatigue compared to sham, 2) Pre-HCT physical function will be a significant predictor of 6-month recovery of physical function and patient-reported outcomes (PRO), and 3) NMES administration will be feasible and well-accepted in the acute HCT setting.

## Materials and methods

### Participants and study design

This protocol was approved by the VA Puget Sound Health Care System (VAPSHCS) Institutional Review Board (IRB; project number 01879, v3 9/14/23 as of date of this submission) and the Research and Development Committee and was conducted in compliance with the Declarations of Helsinki and its amendments and the International Conference on Harmonization Guideline for Good Clinical Practices. It is registered with clinicaltrials.gov (NCT04364256). Protocol modifications which may impact conduct of the study, potential participant benefit or safety, including change of study objectives, design, population, sample size, procedures, or significant administrative aspects will require a formal protocol amendment. Such amendment will be agreed upon by the project Co-Investigators and Data Monitoring Committee (DMC) and approved by the Institutional Review Board prior to implementation and notified to the study sponsor. Minor administrative corrections and/or clarifications with little effect on study conduct will be agreed upon by the project Co-Investigators and will be documented in a memorandum. The Ethics Committee/IRB may be notified of administrative changes at the discretion of the project Co-Investigators.

Subject recruitment from the Bone Marrow Transplant Unit (MTU) at the VAPSHCS in Seattle, WA, was initiated July 1, 2020, and is anticipated to continue through December 31, 2024. When a patient is being evaluated for enrolment into the MTU, clinical staff at the MTU

| Parameter | | Study Procedures | Enrolment | PRE | FU1 | FU2 |
|---|---|---|---|---|---|---|
| Enrolment | | Eligibility screen, Informed consent | X | | | |
| Allocation | | Complete PRE visit and verify soreness criteria | | X | | |
| Intervention | | NMES or sham | | ←——————→ | | |
| Assessments | | | | | | |
| Physical Function | | 1°= 6MWT (manual measurement) | | X | X | |
| | | 2°= 6MWT (mobile measurement) | | X | X | X |
| | | 2°= physical activity (accelerometry) | | X | X | X |
| | | 2°= SCPᵃ, 1-RMᵃ, VO₂peakᵃ, HGS, SPPB (STS, usual gait speed, balance), MVIC (knee flexion/extension) | | X | X | |
| Body Composition | | 2°= Body weight, appendicular LBM, fat mass, fat-free mass | | X | X | |
| Clinical Labs | | Nutrition markers (pre-albumin, albumin, protein), renal function (creatinine), liver function (AST, ALT), hematological parameters (leucocytes, thrombocytes, hematocrit, hemoglobinᵇ, erythrocytes) | | X | X | X |
| Safety Parameters | | Muscle damage (CK), NMES acceptabilityᶜ, AE, survival | | X | X | |
| Patient-Reported Outcomes | | Questionnaires: ASAS, MJM; FACIT-F, SF-36, MFI-20 | | X | X | X |
| Chart Review | | Demographics, vitals, diagnosis, co-morbidities, AE, hospitalizations, medications, transfusions, functional status (i.e., performance rating, C-reactive protein), edema | | X | X | X |
| NMES Process Measures | | Acceptabilityᶜ, adherence, duration, intensity, AE (related to NMES) | | X | X | |

**Fig 1. Schedule of study events and assessments.** [a] Temporarily not assessed due to COVID-19 precautions; [b] measured as a physiologic fatigue parameter; [c] Acceptability & Feasibility Worksheet. FU, follow-up; 1/2⁰, primary/secondary; 6MWT, 6-minute walk test; SCP, stair climb power; 1-RM, 1-repetition maximum; VO₂, aerobic capacity; HGS, handgrip strength; SPPB, short physical function battery; STS, 5-times-sit-to-stand; MVIC, maximal voluntary isometric contraction; LBM, lean body mass; AST, aspartate transaminase; ALT, alanine transaminase; CK, creatine kinase; NMES, neuromuscular electrical stimulation; AE, adverse event(s); ASAS, Anderson Symptom Assessment Scale; FACIT-F, Functional Assessment of Chronic Illness Therapy-Fatigue; MJM, Muscle and Joint Measure; SF-36, Short Form-36 Health Survey; MFI-20, Multidimensional Fatigue Inventory.

will introduce the study to the patient and ask if they are interested in learning more. If they agree, then research staff will be made aware that the patient is amenable to being approached at a clinical visit and will proceed with ethical procedures of obtaining informed consent (Appendix I). As described in Fig 1, measurements of physical function, body composition, and QOL are first assessed Pre-HCT (after enrolment in VAPSHCS MTU and before initiation of preparatory chemotherapy) with a battery of physiological, biomarker, PRO, and clinical measures. These measures are repeated at the primary outcome follow-up at 24 ± 5 days after HCT (FU1). Physical function, PRO (including QOL and fatigue), and clinical outcomes (recurrence, hospitalizations, etc.) also are assessed 6-months after HCT (FU2).

Eligibility criteria include age >18 years, adequate cognitive and language ability to provide consent, and enroled in the VAPSHCS MTU for planned standard of care for autologous HCT. Participants are excluded for history of HCT, active deep vein thrombosis or thrombo-phlebitis, untreated hemorrhagic disorders, probable or definitive liver cirrhosis, reduced renal clearance defined as Stage 3b chronic kidney disease (glomerular filtration rate <45 ml/min/1.73m²), rhabdomyolysis or other muscle conditions where NMES is contraindicated, participation in nutritional or physical exercise interventional trials, concomitant use of anabolic agents, implanted cardiac device, or resting patient-reported muscle soreness of 5–6 on the soreness Likert scale. Participants are not exempt from involvement in regular physiotherapy at the discretion of their treating physician(s); all physiotherapy between Pre and FU2 are recorded.

Participants will receive a $50 compensation payment 1) at the completion of Pre, 2) after completion of FU1, and 3) after study staff receive completed PRO questionnaires for FU2 for a total possible monetary compensation of $150. If a participant is injured as a result of participation in this study, the VA will provide the necessary medical treatment. Participants will not be charged for the necessary medical treatment. Veterans who are injured because of being in this study may receive payment under Title 38, United States Code, Section 1151. Veterans or non-Veterans who are injured may receive payment under the Federal Tort Claims Act. The

VA is not obligated to reimburse medical expenses due to non-compliance with study procedures described in the informed consent or otherwise communicated to participants by study personnel.

## Randomization and blinding

After the Pre-HCT visit, participants are randomized 1:1 with a computerized random number generator to NMES or sham, stratified by diagnosis (multiple myeloma vs. other diagnosis) and sex (male vs. female). All study personnel except the study coordinators are blinded to randomization unless unblinding is required for safety reasons determined by Institutional Review Board or the independent DMC, or until analysis is complete. Study coordinators are unblinded due to their roles in training participants to use the NMES/sham devices and performing weekly check-ins (see "Intervention" section below for details).

Participants are informed that two different signals are being tested in this study: one will be actively felt throughout each session, and one will only be actively felt for the first two minutes of each. This strategy was chosen to remain transparent about the intervention arms while minimizing likelihood of patient unblinding, which would be extremely high if informed of an 'active' and a 'sham' group. However, subjects are also informed that we do not expect every outcome to be affected in both groups, and that the primary study goal is to determine which outcomes are affected, if any, in each group.

## Intervention

The intervention will be initiated after randomization, with the first session serving as a training session under the supervision of the unblinded study coordinator where participants receive instruction on safety, muscle stimulator operation, and electrode placement and become acquainted with the stimulus sensation. All sessions thereafter will be unsupervised unless the participant requests follow-up guidance/clarification. The intervention will consist of three sessions/week and will continue until three weeks post-HCT. Each intervention session will be one hour total: 30 minutes of simultaneous bilateral NMES/sham of the gluteals and hamstrings and 30 minutes of bilateral NMES/sham of the quadriceps. NMES is delivered by the RS-4i Plus Sequential Stimulator (RS Medical, Vancouver, WA) using asymmetric biphasic waveforms at 71 pps (Hz), 0–400 μs pulse duration, 5:10s on:off time (33% duty cycle), and 1.5s ramp-up. Participants control the muscle stimulator at all times and are instructed to perform all sessions in the supine position. Bilateral NMES is delivered using four cutaneous parallel channels delivered simultaneously using 2"x4" or 3"x5" self-adhesive electrodes (Fig 2). For the NMES group, participants are encouraged to increase the amplitude to a level of moderate discomfort, such as experienced during conventional exercise, but not to induce pain. At minimum, the amplitude should induce visible muscle contraction. The sham group uses the same stimulator and electrode placement, and the first two minutes of stimulation are the same as the active NMES parameters with the patient being allowed to increase the amplitude as tolerated. However, the amplitude will automatically ramp down to 0 milliamps over the third minute and stay there for the remaining 28 minutes. During this time, the patient can observe the amplitude bar increasing on the device screen if they push the "up" button, but the output remains 0. Sham was chosen over usual care or transcutaneous electrical stimulation comparator groups to maximize NMES simulation without major risk of inducing confounding biological effects such as pain reduction. On each intervention day, an unblinded study coordinator calls the patient to 1) assess for adverse events (AE), including muscle soreness and lab abnormalities, 2) discuss issues related to electrode placement, 3) check for

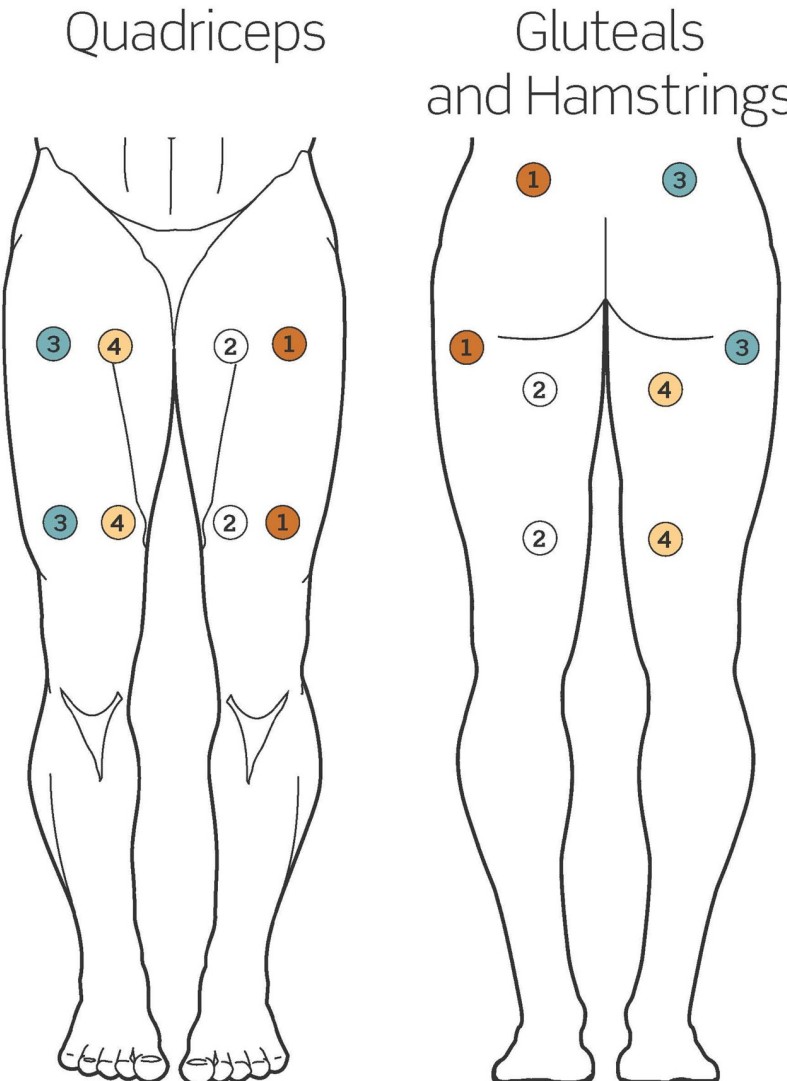

**Fig 2. Electrode placement guide.** Each circle represents one electrode (actual electrode size varies), electrodes are connected in pairs or "channels" according to matching number/color within each circle. Stimulation of the quadriceps involves two channels per leg while stimulation of gluteals and hamstrings involves one channel on gluteals and one channel on hamstrings per side. This placement guide was developed for this study with the assistance of the device manufacturer and is being printed here with their explicit permission (RS Medical, Vancouver, WA, USA).

changes in skin color, moisture, integrity and/or sensitivity, 4) ask about engagement in other physical activities, and 5) provide general adherence encouragement.

Criteria for withholding/restarting the intervention: Intervention will be held in the case of patient-reported soreness of 5–6 on a 0–6 likert scale. The study coordinator will ask the subject about the time course of soreness development and about any signs of systemic muscle damage like darker than normal urine color. This information will be relayed to the study physician(s) for their determination of any necessary clinical follow-up. At minimum, soreness will be measured every 24 hours and the intervention may begin again when soreness falls between 0–4. Intervention will also be held in the case of Grade 3 (5-10x upper limit of normal) or higher CK elevation independent of urinalysis or muscle soreness. CK will then be

measured every 48 hours and the intervention may begin again when CK drops below Grade 3. This precaution will be re-assessed after 5–10 patients have completed the intervention to determine the frequency of CK elevations and associated symptoms. If Grade 3 occurs frequently without urine or muscle symptoms, then the CK elevation criterion to hold the intervention may be raised per study physician recommendation.

## Outcome measures

The primary outcome is the between-group difference in 6MWT change (FU1 minus Pre-HCT). Despite that the 6MWT is conventionally used as a sub-maximal endurance test, high frequency-NMES, such as that utilized here, has repeatedly improved 6MWT performance, and therefore is used as an indicator of global physical function, in chronic disease cohorts including those with advanced cancer [17,24,28–30]. Secondary measures include physical function assessed at Pre-HCT and FU1 by HGS, Short Physical Performance Battery, maximal voluntary isometric contraction of quadriceps and hamstrings, and physical activity. Physical function at FU2 consists of 6MWT and physical activity. Body composition is measured by dual-energy x-ray absorptiometry (DXA) and bioelectrical impedance analysis at Pre-HCT and FU1 only. PROs are measured via questionnaires to assess QOL, fatigue, symptom burden, and functional status at Pre-HCT, FU1, and FU2. Additional data collected to describe the sample and consider meaningful modifiers of outcomes include clinical and laboratory data acquired per standard of care and process measures such as feasibility, acceptability, and adherence. We have previously reported that 6MWT, HGS, and knee flexor/extensor strength are sensitive to capture changes in physical function acutely post-HCT [31].

## Physical function

**6-Minute walk test (primary outcome).** Participants are instructed to walk as quickly as possible, without running, back and forth down an open hallway, across a distance of 20 meters. Participants may stop as needed, but the timer will not pause; the total distance walked is recorded after six continuous minutes. At Pre-HCT and FU1, 6MWT is monitored and manually recorded by study staff and by a mobile phone-based application called iWalkAssess which is commercially available on both android and iOS platforms [32]. At FU2, 6MWT is only measured by iWalkAssess; patients are mailed a 20-meter string for assistance. The VAPSHCS MTU is one of three MTU in the VA system, so the majority of study participants reside out of state and must complete FU2 remotely.

**Hand grip strength (HGS).** Participants squeeze a handheld dynamometer (Jamar Hydraulic Dynamometer, J.A. Preston Corp., Clifton, NJ) as hard as possible for five seconds [33]. Three attempts are made on each hand, with the highest measure per hand recorded and averaged for reporting mean HGS.

**Short physical performance battery.** This battery includes assessment of 1) 5-times-sit-to-stand: Participants begin seated, then are timed while standing up and sitting down five times as quickly as possible, maintaining their arms crossed on their chest. This test may be performed up to three times to ensure test familiarization and the best performance is recorded for analysis; 2) Gait speed: Participants walk at their usual pace for four meters, from a standing start. The time employed to walk is used to calculate gait speed in meters/second and recorded for analysis; and 3) Balance: Tests of standing balance include tandem, semi-tandem, and side-by-side stands. Each participant begins with the semi-tandem stand, in which the heel of one foot is placed to the side of the first toe of the other foot, with the participant choosing which foot to place forward. Those unable to hold the semi-tandem position for 10 seconds are evaluated with the feet in the side-by-side position. Those able to maintain the

semi-tandem position for 10 seconds are further evaluated with the feet in full tandem position, with the heel of one foot directly in front of the toes of the other foot. The timing is stopped when the participant moves their feet or reaches out for support, or when 10 seconds has elapsed. The Short Physical Performance Battery is feasible in patients with hematologic cancer undergoing treatment [34].

**Maximal voluntary isometric contraction.** Maximal strength of the knee extensors and flexors is assessed using handheld dynamometry (MicroFET 2; Performance Health, Warrenville, IL) where patients push or pull against a foam-padded hand-held force transducer for five seconds to assess their maximum isometric force. Participants undergo 1–2 practice/warm-up attempts and then 2–3 maximal attempts, with one minute rest in between [35]. This dynamometer has been previously used to assess maximal voluntary isometric contraction in the HCT setting [36].

**Physical activity.** Validated triaxial accelerometry (Actical, Philips Respironics, Murrysville, PA) is used to determine daily spontaneous physical activity [37]. Actical is worn by patients at the wrist for seven days at Pre-HCT, FU1, and FU2. Data from returned monitors is downloaded with companion software and analyzed for activity counts, energy expenditure, and daily steps.

## Patient-reported outcomes

Fatigue and QOL are measured by the Anderson Symptom Assessment Scale (ASAS), Muscle and Joint Measure (MJM), FACIT-F, Short Form-36 Health Survey (SF-36), and the MFI-20. The ASAS has been previously validated in the palliative care setting [38] and is widely used to capture symptom distress in cancer populations [39–41]. Patients rate 10 symptom categories from 0 (indicating lack of burden or no issues) to 10 (indicating "worst imaginable" burden). Individual category ratings are flipped by study staff during scoring and then summed together for a cumulative "ASAS Total" score (range: 0–100) to be used for reporting. Using this method, larger "ASAS Total" scores represent better QOL and increases in "ASAS Total" score over time indicate QOL improvement. The MJM measures musculoskeletal symptoms in HCT survivors [42] and is proven sensitive enough to detect symptom improvements in this setting [43]. The questions in this survey may direct respondents to rate severity numerically or they may be qualitative with multiple choice answers where each answer is pre-assigned a numerical value for scoring. Symptom severity is then summed for reporting in four categories: Muscle Cramps and Spasms, Muscle Weakness, Myalgia, and Arthralgia; the number of questions summed for each category is symptom dependent. Larger scores indicate greater symptom severity and increased score over time indicates symptom worsening. The FACIT-F has been widely used in cancer-related fatigue studies, can detect clinically meaningful differences in fatigue scores, and was shown to improve after NMES [24]. Participants are asked to rate their agreement with 40 statements ranging from 0 ("not at all") to 4 ("very much"). For some statements a 4-rating would indicate good QOL and for others, a 4-rating would indicate bad QOL; therefore, some individual items require flipping for scoring consistency. QOL is reported in five categories: Physical Well-Being (range: 0–28; all flipped), Social/Family Well-Being (range: 0–28; none flipped), Emotional Well-Being (range: 0–24; 5 are flipped), Functional Well-Being (range: 0–28; none flipped), and Fatigue (range: 0–52; 11 are flipped). These categories are also summed together for reporting "FACIT-F Total" (range: 0–160). Larger category or total scores represent better QOL and increases in category or total scores over time indicate QOL improvement. The MFI-20 assesses fatigue across five categories (range: 4–20 each) including General Fatigue, Physical Fatigue, Reduced Activity, Reduced Motivation, and Mental Fatigue with ability todetect clinically important differences [27,44,45]. It has also shown to improve

after NMES [23]. Participants rate their agreement with 20 statements from 1 (true) to 5 (not true) and the total for each category is summed for reporting. The standardized instructions for this questionnaire also require the flipping of specific items for scoring; however, in contrast to the other parameters described here, the MFI flipping procedure results in larger scores indicating greater fatigue severity and increases over time indicate fatigue exacerbation. The SF-36 is a validated instrument for measuring health perception [46], can detect clinically important differences in fatigue [47], and has been used to assess QOL in the HCT population [48]. We have previously reported that the FACIT-F and SF-36 are sensitive to capture changes in patients undergoing HCT [31]. Participants are asked multiple-choice questions about frequency and magnitude of symptom burden with each answer pre-assigned a numerical value for scoring. Items are summed for reporting in eight categories: General Health Perception (score range: 0–20), Physical Functioning (score range: 0–20), Role Limitations due to Physical Problems (score range: 0–4), Role Limitations due to Emotional Problems (score range: 0–3), Social Functioning (score range: 0–8), Bodily Pain (score range: 0–9), Vitality (score range: 0–20), and Mental Health (score range: 0–20). Larger category or total scores represent better QOL and increases in category or total scores over time indicate QOL improvement.

## Body composition

After an overnight fast, DXA (Hologic Inc., Marlborough, MA) and bioelectrical impedance analysis (InBody 770, BioSpace, Cerritos, CA) are measured to estimate total lean body mass, appendicular lean body mass, and total fat mass [49,50]. Participants dress in a hospital gown and undergarments for the DXA scan.

## NMES feasibility and acceptability

Participants complete an acceptability and feasibility form at FU1 (Acceptability & Feasibility Worksheet). In addition, the NMES device has an internal function where users rate their pain before and after each NMES session which is also collected as part of acceptability and adherence. As a strategy to minimize missing data, any patient that elects to withdraw from the study will be informed of the option to only withdraw from the intervention and remain enroled for completion of FU1/2.

## Safety

**Laboratory parameters.** Venipuncture is performed at Pre-HCT and FU1 to measure muscle damage (CK), CK is also measured every two weeks after initiation of NMES/sham; remaining plasma is kept in storage at -80˚C for future research related to these objectives unless patients sign an independent consent form for a VAPSHCS biospecimen repository. Laboratory values measured per standard of care include nutritional markers, renal function, liver function, and hematological parameters in plasma at Pre-HCT and FU1 (Fig 1); hemoglobin is also obtained as a physiologic fatigue parameter.

**Adverse event reporting.** Other safety parameters include AE (National Cancer Institute Common Terminology Criteria for Adverse Events v5.0), Acceptability & Feasibility Worksheet (assessed at FU1), and survival (assessed at FU2). All "unanticipated and related SAE" and "unanticipated and related problem involving risk to subjects or others" will be reported to IRB with appropriate forms per IRB regulations (within 5 business days of the reporting individual becoming aware of the event), all other AEs will be reported to IRB annually. The IRB director will be notified orally of all unanticipated related deaths immediately after the reporting individual becoming aware of the event. The device manufacturer, RS Medical, will also be notified of all unanticipated and related SAEs within five business days of the reporting

individual becoming aware of the event; RS Medical will be responsible for reporting these SAEs to the FDA. Exclusion criteria are based on established contraindications to NMES [51]. We do not expect significant differences in NMES-related AE (with the possible exception of muscle soreness) as previously reported [34]. Muscle damage (CK) may display clinically insignificant increases within the first week of NMES (asymptomatic Grade 1–3 elevations) with levels expected to return to Pre-HCT within 1–2 weeks.

## Outcomes proposed but unlikely to be assessed due to COVID-19 precautions

**Peak aerobic capacity.** After an overnight fast, and under direct supervision of a study physician, participants wear a mask/mouthpiece with a breathing valve to collect expired gases via indirect calorimetry while pedaling on a cycle ergometer at progressively harder workloads. The test is a ramping protocol starting at 0 watts with a continuously increasing workload of 15, 20, or 25 watts/minute and continues until the participant becomes fatigued and decides to stop, cannot maintain 65rpm, and/or other symptoms prohibit further exercise. Participants rate their highest perceived exertion from the ramping test using the Borg Scale ranging from 6 to 20 [52] during a cool-down exercise where they continue to cycle at a self-selected pace with zero resistance on the ergometer for 3–5 minutes. Heart rate is monitored throughout the test. Peak oxygen consumption rate (ml/kg/min) prior to cool-down is recorded for analysis.

**Stair climb power.** Participants ascend a flight of stairs at the highest possible speed, according to their capabilities. The stairs consist of 13 steps of 15.3 cm each, thus covering a total vertical distance of 1.99m. The study coordinator measures the time employed to complete the test with a digital stopwatch. Anaerobic power (in Watts) is calculated as follows: "[body mass (kg) x 9.81 (m/s$^2$) x vertical distance (m)] / time (s)" where 9.81 m/s$^2$ represents the acceleration of gravity. Two or three practice trials are allowed so participants gain good control of the technique, with shortest time to completion used for data analysis.

**1-Repetition maximum.** Participants will complete maximal strength testing of three major lower body muscle groups on pneumatic exercise equipment (Keiser Corp., Fresno, CA): quadriceps (bilateral knee extension), hamstrings (bilateral knee flexion), and gluteals (unilateral hip extension). Participants are fitted to each exercise machine in accordance with proper body mechanics. Before initiating any exercise, participants are instructed on the strength protocol, proper form, and breathing technique and are given reminder cues for proper form and breathing as needed throughout each exercise. Strength testing begins with assessment of range of motion performed without any resistance, allowing participants to get familiar with the appropriate start/stop points. Two warm-up sets are then performed with resistance (1: 7–8 repetitions at approximately 5/10 difficulty, 2: 4–5 repetitions at approximately 7/10 difficulty) to increase circulation and neural recruitment. Next, participants begin one-repetition maximal strength attempts, with one-minute of rest in-between, with the goal of achieving maximal effort within 3–5 attempts. Maximal strength is recorded from the last successful attempt while maintaining proper form and range of motion.

## Statistical analysis and sample size calculation

The primary outcome is between-group difference in 6MWT change from Pre-HCT to FU1 (manual assessment). Our laboratory previously reported ~41 ± 15m reduction in 6MWT in Autologous recipients between the same Pre and FU1 time points [31]. We expect the sham group in the current project to display a similar change, and for NMES to attenuate this change by 50%, resulting in a between-group difference in 6MWT change of -20.7 ± 16.0m. Along with a Type I error probability α = 0.05, power = 0.9, and 1:1 NMES:sham participant

randomization, we will need to enrol 15 minimum participants per group. This is consistent with the recent finding from Bewarder and colleagues in which a combined group of autologous/allogenic HCT patients and hematologic cancer patients undergoing intensive chemotherapy without HCT displayed a mean reduction in 6MWT of 24m after NMES [25]. We estimate a 33% attrition rate, and therefore aim to enrol 23 participants/group (46 total). Approximately 100 HCT are performed each year at the MTU at VAPSHCS with roughly 70 being autologous HCT. The enrolment rate for our prior observational trial was 20 autologous HCT recipients/year, and we estimate enrolment of 17 patients/year for this project which would allow for completion of enrolment (N = 46 total), accounting for attrition, and primary outcome (FU1) assessment 2.5–3 years after study initiation and completion of final (FU2) assessment 3–3.5 years after study initiation.

All variables will be summarized descriptively using N, mean, and standard error using the latest version of SPSS. The study cohort will be described using Pre-HCT data, both overall and comparing between study arms using t-tests for continuous variables and $X^2$ tests for categorical variables. Testing of statistical significance will be 2-tailed, with a p-value ≤0.05. An intent-to-treat analyses will be conducted using paired-sample t-tests for continuous variables and $X^2$ for categorical variables for within-group comparisons of outcomes between Pre and FU1; independent t-tests will be used for between-group comparisons of change variables (FU1 minus Pre-HCT). Exploratory analyses will include multiple linear regression models to test whether Pre-HCT 6MWT is a predictor of physical function or patient-reported fatigue (FACIT-F or MJM) at FU2. The model will control for group assignment, adherence, hemoglobin, and age as well as any potential confounding variables found to be unbalanced between groups in bivariate testing described above.

## Data management and access plan

All VA data is maintained on secure VA servers or, if hard copy, in secured facilities behind double lock. Investigators and staff are trained annually regarding data security standards. Policies, procedures and mechanisms are in place to ensure compliance with Federal and State requirements for the protection of privacy and confidentiality, Personally Identifiable Information, and Protected Health Information. Final data sets underlying publications resulting from this research will be made available outside VA in response to a properly prepared Freedom of Information Act request submitted to the VAPSHCS FOIA Officer or submitted and passed down to the facility FOIA Officer from higher VA authority. Such requests will be evaluated for protection of potential intellectual property value and rights of VA and its employees, collaborators, research sponsors, and any other proprietary or security concerns including protecting Protected Health Information & Personally Identifiable Information. Sharing of data is germane to advancement of science/medicine for public good. Sharing of data, if made available upon approval of a FOIA request, may enable the recipient to design studies to test the results originally obtained by the PI who generated the data or to expand upon the work. This assumes the recipient has sufficient knowledge, training, and resources to adequately design and conduct the studies replicating the original work or can otherwise determine results validity by data review. Publications from this research will be made available to the public through the National Library of Medicine PubMed Central website within one year after the date of publication.

## Data monitoring committee

A Data DMC will conduct a blinded examination of the safety data to ensure safety of subjects. All study AE will be documented and tabulated for DMC reporting. Unanticipated and related SAEs will be reported to the DMC within five business days of the investigators becoming aware

of the event. The DMC will meet to review safety reports every six months. Each safety report will include the enrolment status and number of subjects recruited, screened, and randomized. In addition, it will incorporate the number of dropouts and the reason for dropouts. The report will list all AE and SAE that did not require expedited reporting since the previous reporting period. The summary of all AE will allow for a comprehensive overview of the event rates. All AE will be evaluated for severity and attribution. In order to protect participant's confidentiality, the data provided to the DMC will be coded so that individual patients cannot be identified.

## Results and discussion

### Expected primary outcome results

The sham group is expected to display a reduction in 6MWT of approximately 40m, as previously reported from our lab in an observational study [31], and for the NMES intervention to attenuate this change by 50%, or 20m, for the NMES group. This would be consistent with the recent NMES feasibility study in patients with hematologic cancer [34]. If we observe <50% attenuation or a statistically insignificant attenuation, we will determine whether the impact of NMES is clinically meaningful. As the minimal clinically important difference for the 6MWT is 30.5m [26], a clinically meaningful impact of NMES on physical function would be indicated by a decrease in <30.5m for the NMES group or a mean increase of >30.5m for the sham group. Alternatively, a clinically meaningful impact of NMES could also be indicated by a between-group difference in 6MWT change score of >30.5m. We also anticipate the NMES group will display a maintenance of 6MWT (mobile assessment) from FU1 to FU2 and from Pre-HCT to FU2, compared to sham, which will display a statistically significant worsening. Pre-HCT 6MWT will be used to determine significant predictors of physical function, muscle mass, and QOL, and fatigue at FU1 and FU2. Exploratory measures of physical function (HGS; Short Physical Performance Battery; muscle strength; physical activity) assessed at Pre-HCT and FU1 will be used to determine significant predictors of QOL at FU2 for determining potential endpoints for future clinical trials.

### Potential limitations

Maintenance of participant blinding may be difficult due to the nature of the intervention being a muscle stimulator, accessibility of information available online, and patient interaction at the MTU. Participants may become aware that they are receiving the sham stimulus. An alternative strategy will be to ask the participants at the end of the study if they knew of their group assignment and do a sensitivity analysis to see if this knowledge biased any subjective outcomes. Another issue to account for is that DXA methodology includes body water as part of lean mass measurements which may be influenced by variable edema of patients undergoing complex treatment, especially with exposure to high-dose corticosteroids. We will assess the patients for edema per chart review (trace, 1+, 2+, 3+ anasarca) and perform a sensitivity analysis. Concurrent engagement in physiotherapy may be another confounding variable; however, the current rate of consultation to physical therapy for patients enroled at the MTU at VAPSHCS is 10% so we will not exclude concurrent physiotherapy and do not anticipate this to be a significant confounder. In addition, the lack of ability to control for nutrition, particularly during the post-intervention period, may contribute to physical function and quality of life at FU2, and could be an important factor to be addressed in future studies. Also, this trial was only statistically powered for the primary outcome and may be under-powered for remaining outcomes, which will need to be considered during final data analysis. Finally, external validation will be required for any observations reported from these proposed analyses to ensure confidence in generalizability to the broader patient population.

## Conclusion

Hematopoietic stem cell transplantation is a common strategy for those with hematologic malignancies but can be associated with deconditioning, fatigue, sarcopenia, and poor QOL. Despite the significance of these symptoms, there are no approved treatments to prevent or limit these long-term adverse outcomes. Traditional exercise regimens implemented before and/or after HCT have helped reduce these effects, but exercise is inconsistently recommended to and practiced by HCT patients. NMES improves physical function, muscle mass, and fatigue in cohorts with similar functional impairments as HCT recipients. The use of NMES to combat sarcopenia and functional decline may be particularly useful in the HCT setting as patients undergo intensive preparatory chemotherapy and often experience severe fatigue and inactivity for extended periods. This first randomized-controlled trial of NMES in the HCT setting is of great importance considering there are no currently approved treatments for functional impairment, muscle atrophy, or fatigue in patients undergoing HCT or in the broader cancer setting. The study will contribute to developing strategies toward optimizing the functional and QOL outcomes for patients undergoing HCT.

## Supporting information

**S1 File.**
(PDF)

**S2 File.**
(PDF)

## Acknowledgments

This study was supported by resources and facilities at the VAPSHCS Geriatric Research, Education and Clinical Center. RS Medical contributed the RS-4i Plus Sequential Stimulators, were involved in design of the NMES and sham parameters and provided technical support throughout the study.

## Author Contributions

**Conceptualization:** Lindsey J. Anderson, Karen L. Syrjala, Thomas R. Chauncey, Jose M. Garcia.

**Data curation:** Gary Miranda.

**Funding acquisition:** Lindsey J. Anderson.

**Investigation:** Lindsey J. Anderson, Lauren Paulsen, Gary Miranda.

**Methodology:** Lindsey J. Anderson, Karen L. Syrjala, Solomon A. Graf, Thomas R. Chauncey, Jose M. Garcia.

**Project administration:** Lindsey J. Anderson, Lauren Paulsen.

**Resources:** Solomon A. Graf, Thomas R. Chauncey, Jose M. Garcia.

**Supervision:** Solomon A. Graf, Thomas R. Chauncey, Jose M. Garcia.

**Writing – original draft:** Lindsey J. Anderson, Lauren Paulsen, Gary Miranda.

**Writing – review & editing:** Lindsey J. Anderson, Karen L. Syrjala, Solomon A. Graf, Thomas R. Chauncey, Jose M. Garcia.

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
