## [Decision Letter · Decision Letter 0]

18 Dec 2023

PONE-D-23-33264Neuromuscular electrical stimulation for physical function maintenance during hematopoietic stem cell transplant: Study ProtocolPLOS ONE

Dear Dr. Anderson,

Thank you for submitting your manuscript to PLOS ONE. After careful consideration, we feel that it has merit but does not fully meet PLOS ONE’s publication criteria as it currently stands. Therefore, we invite you to submit a revised version of the manuscript that addresses the points raised during the review process.

We look forward to receiving your revised manuscript.

Kind regards,

Masoud Rahmati

Academic Editor

PLOS ONE

Journal Requirements:

"This study was funded by a grant from the U.S. Dept. of VA (grant number RX003245)"

'This study was supported by resources and facilities at the VAPSHCS Geriatric Research, Education and Clinical Center and JMG receives research support from the VA (BX002807), the Congressionally Directed Medical Research Program (PC170059), and the National Institutes of Health (NIH; R01CA239208, R01AG061558). LJA receives research support from the University of Washington (DK007247) and the VA (RX003245). RS Medical contributed the RS-4i Plus Sequential Stimulators, were involved in design of the NMES and sham parameters, and provided technical support throughout the study."

"This study was funded by a grant from the U.S. Dept. of VA (grant number RX003245)"

6. We note that the original protocol file you uploaded contains a confidentiality notice indicating that the protocol may not be shared publicly or be published. Please note, however, that the PLOS Editorial Policy requires that the original protocol be published alongside your manuscript in the event of acceptance. Please note that should your paper be accepted, all content including the protocol will be published under the Creative Commons Attribution (CC BY) 4.0 license, which means that it will be freely available online, and any third party is permitted to access, download, copy, distribute, and use these materials in any way, even commercially, with proper attribution.

Therefore, we ask that you please seek permission from the study sponsor or body imposing the restriction on sharing this document to publish this protocol under CC BY 4.0 if your work is accepted. We kindly ask that you upload a formal statement signed by an institutional representative clarifying whether you will be able to comply with this policy. Additionally, please upload a clean copy of the protocol with the confidentiality notice (and any copyrighted institutional logos or signatures) removed.

Reviewers' comments:

Reviewer's Responses to Questions

**Comments to the Author**

1. Does the manuscript provide a valid rationale for the proposed study, with clearly identified and justified research questions?

Reviewer #1: Yes

Reviewer #2: Yes

2. Is the protocol technically sound and planned in a manner that will lead to a meaningful outcome and allow testing the stated hypotheses?

Reviewer #1: Partly

Reviewer #2: Yes

3. Is the methodology feasible and described in sufficient detail to allow the work to be replicable?

Reviewer #1: Yes

Reviewer #2: Yes

4. Have the authors described where all data underlying the findings will be made available when the study is complete?

Reviewer #1: Yes

Reviewer #2: Yes

5. Is the manuscript presented in an intelligible fashion and written in standard English?

Reviewer #1: Yes

Reviewer #2: Yes

6. Review Comments to the Author

You may also provide optional suggestions and comments to authors that they might find helpful in planning their study.

Reviewer #1: In this study protocol, the authors investigated the efficacy of NMES for mitigating the detrimental effect of hematopoietic stem cell transplant in patients with cancer. The application of NMES is patients with cancer is emerging and with a promising evidence. Therefore, new studies are welcome to have a comprehensive understanding of the efficacy and drawbacks of this strategy. I have included some comments for authors’ consideration as follows:

1. Line 71-74; The sentence needs correction.

2. Line 66-79 (Paragraph 2); In order to shorten the introduction, the information provided in the second paragraph could be summarized. Moreover, the structure and the information provided in the second paragraph of the introduction are as if exercise is the main intervention in this study while the authors just want to justify the use of NMES instead of exercise training. So, I recommend to re-structure the second paragraph and remove unnecessary information.

3. In the introduction, the authors have appropriately presented the current knowledge and understanding concerning the application of NMES in clinical populations such as cancer, however, it would be more interesting if the authors briefly highlight which specific gaps in the literature are covered in this study and what is the main novelty of this project.

4. I congratulate the authors for providing the detailed information in the “Participants and study design” section.

5. Line 173; No information regarding the randomization and blinding procedure have been provided. How did the author evaluate the efficacy of the blinding strategy used in this study?

6. Line 174; Since the replication of a study protocol by other researchers is important, I recommend the authors to provide more information regarding the “stratification by diagnosis”.

7. Could the authors elaborate more on their choice of the NMES parameters? Considering the fact that the main outcome measure of this study is 6MWT and that low frequency-NMES is more likely to induce aerobic adaptations while high frequency-NMES is proven to have strengthening effect on target muscles.

8. Line 255, Please provide a reference for using one-minute rest between MVICs.

9. Line 261; Instead of providing justifications for the choice of instruments used for “Patient-Reported Outcomes”, I recommend the authors to provide detailed information regarding the scoring and interpretation of these measures.

10. Line 304; Please provide more information regarding the incremental test on the ergometer for measuring peak aerobic capacity. What was the initial workload? How much load was added at each stage and what was the duration of each stage? Was the verbal encouragement provided? When the participants reported their RPE and by which Borg scale?

11. Line 402; Did the authors have any control on nutritional status of the participants during the project (in particular between FU1 and FU2)? If so, please provide the related information, if not, it could be another limitation of the study.

12. Line 403-407; I addressed this issue in my 5th comment. I think it is necessary that the author provide more information regarding any strategies used for blinding in the related section of the manuscript. Nevertheless, I agree that the nature of NMES makes it difficult to blind the participants.

Reviewer #2: As the statistical reviewer I will focus on methods and reporting. This is a well planned and reported study. However, I have some comments to make.

1) the term pilot is used at times. As far as I understand, this relates to very small trials that obtain an effect estimate to be used for a power calculations towards designing a larger trial. Some clarification is needed, as to whether that is the case here, or the term trial needs to be dropped.

2) the authors estimate an attrition rate of 50% and they need 30 participants, but they aim to recruit 46 rather than 60 - is there a mistake somewhere?

3) power may be inadequate for other outcomes, so this needs to be discussed as a limitation and needs to be reported when the authors conduct their analyses.

4) randomisation may not work that well for a sample this small, so maybe the authors could consider more deterministic approaches, for example please see https://www.jstatsoft.org/article/view/v055c01. They may want to ignore this advice, but this would be my preferred approach. Randomisation may work fine, by chance.

5) and regression models will not work well with 30 subjects, so you may want to control for numerous covariates where imbalances exist, but in practice you will struggle to include more than a handful of covariates, especially if categorical. So it's better to strive for balance, for at least a few key covariates using a deterministic algorithm in my view, although the size of the sample means some imbalances will definitely exist.

7. PLOS authors have the option to publish the peer review history of their article (what does this mean?). If published, this will include your full peer review and any attached files.

Reviewer #1: **Yes: **Ehsan Amiri

Reviewer #2: No

---

## [Author Response · Author response to Decision Letter 0]

20 Mar 2024

The authors thank the journal editorial team and the reviewers for their careful attention and thoughtful feedback which we have responded to point-by-point below. PLEASE NOTE: LINE NUMBERS IN AUTHOR RESPONSES REFER TO THE CLEAN MANUSCRIPT FILE.

Journal Requirements:

Response to journal comment:

The title page components, Abstract, all headings and reference to Figures throughout the manuscript, reducing number of abbreviations, Figure format (including upload to PACE), and file naming have been revised per guidelines provided. Please note, the main body example link provided above indicates Figure legends should be placed “in line” with the Figure label and title; however, the submission guideline site found here Figures | PLOS ONE indicates the legend should be placed “after” the label and title. We have inserted the legend “after” the label and title. In addition, the title page example link provided above does not display a Short Title which is required per the submission site Submission Guidelines | PLOS ONE so we have included one here.

Please find response to journal comments #2-4 provided as one summary response after #4.

3. Thank you for stating the following financial disclosure: "This study was funded by a grant from the U.S. Dept. of VA (grant number RX003245)"

Please find response to journal comments #2-4 provided as one summary response after #4.

'This study was supported by resources and facilities at the VAPSHCS Geriatric Research, Education and Clinical Center and JMG receives research support from the VA (BX002807), the Congressionally Directed Medical Research Program (PC170059), and the National Institutes of Health (NIH; R01CA239208, R01AG061558). LJA receives research support from the University of Washington (DK007247) and the VA (RX003245). RS Medical contributed the RS-4i Plus Sequential Stimulators, were involved in design of the NMES and sham parameters, and provided technical support throughout the study."

We note that you have provided funding information that is not currently declared in your Funding Statement. However, funding information should not appear in the Acknowledgments section or other areas of your manuscript. We will only publish funding information present in the Funding Statement section of the online submission form. Please remove any funding-related text from the manuscript and let us know how you would like to update your Funding Statement. Currently, your Funding Statement reads as follows: "This study was funded by a grant from the U.S. Dept. of VA (grant number RX003245)"

Response to journal comments #2-4:

Comments #2-4 all refer to inconsistency/inaccuracy in the funding, disclosure, and/or acknowledgement sections; all three revised statements are provided below and in the revised Cover Letter. We are responding to all three here for efficiency. 

Direct funding for the research has been removed from Acknowledgements and added to the funding statement. No financial support was received specifically for this research from the sources listed in the current Acknowledgments statement.

The Protocol Article Template found here indicates there should be a funding statement and a competing interest statement on the title page, yet these are not displayed on the sample title page provided in #1 above, and #4 above states you only publish funding information entered into the online submission form. https://journals.plos.org/plosone/s/file?id=c9fb/Study%20Protocol%20Article%20Template.pdf

Please clarify whether the funding and/or competing interest statements are required on the title page.

Funding: This study was funded by a grant from the U.S. Dept. of VA (grant number RX003245; PI: Anderson); the Dept. of VA had no role in study design, data collection and analysis, decision to publish, or preparation of the manuscript. JMG receives research support from the VA (BX002807), the Congressionally Directed Medical Research Program (PC170059), and the National Institutes of Health (NIH; R01CA239208, R01AG061558). 

Competing interests: The authors declare no conflicts of interest

Acknowledgments: This study was supported by resources and facilities at the VAPSHCS Geriatric Research, Education and Clinical Center and JMG receives research support from the VA (BX002807), the Congressionally Directed Medical Research Program (PC170059), and the National Institutes of Health (NIH; R01CA239208, R01AG061558). These supporters had no role in study design, data collection and analysis, decision to publish, or preparation of the manuscript.

Response to journal comment:

We apologize for the confusion on this. The original data statement pertained to data generated from the actual trial, it did not pertain to this submission of the study protocol. The revised statement shown here is also contained in the revised Cover Letter:

Data availability: the data availability policy is not applicable as this study protocol does not report data.

6. We note that the original protocol file you uploaded contains a confidentiality notice indicating that the protocol may not be shared publicly or be published. Please note, however, that the PLOS Editorial Policy requires that the original protocol be published alongside your manuscript in the event of acceptance. Please note that should your paper be accepted, all content including the protocol will be published under the Creative Commons Attribution (CC BY) 4.0 license, which means that it will be freely available online, and any third party is permitted to access, download, copy, distribute, and use these materials in any way, even commercially, with proper attribution.

Therefore, we ask that you please seek permission from the study sponsor or body imposing the restriction on sharing this document to publish this protocol under CC BY 4.0 if your work is accepted. We kindly ask that you upload a formal statement signed by an institutional representative clarifying whether you will be able to comply with this policy. Additionally, please upload a clean copy of the protocol with the confidentiality notice (and any copyrighted institutional logos or signatures) removed.

Response to journal comment:

We thank the journal for their response to our email inquiry asking for further guidance. Regarding copyrighted logos or signatures, thank you for verifying the following “It does appear your protocol is already free of such incompatible elements.” Regarding the Confidentiality Notice, I can confirm that Dr. Patricia Dorn, Director of the VA Rehabilitation R&D service line (study sponsor), has determined that there are no restrictions on publication of the protocol document. I have uploaded an email from Erin Spaniol, Deputy Director of Rehabilitation R&D indicating the following statement:

“We spoke with Dr. Dorn and she is in agreement that the protocol document is your intellectual property and you are able to share at your discretion.”

Upon further clarification from the sponsor, we have also uploaded the consent form with name and phone number of study coordinator redacted. In addition, once enrolment is closed for this active trial, the consent form approved at that time will be published on this trial’s webpage on clinicaltrial.gov as required per VHA Directive 1200.05(3). 

Reviewers' comments:

Reviewer's Responses to Questions

Comments to the Author

1. Does the manuscript provide a valid rationale for the proposed study, with clearly identified and justified research questions?

Reviewer #1: Yes

Reviewer #2: Yes

2. Is the protocol technically sound and planned in a manner that will lead to a meaningful outcome and allow testing the stated hypotheses?

Reviewer #1: Partly

Reviewer #2: Yes

3. Is the methodology feasible and described in sufficient detail to allow the work to be replicable?

Reviewer #1: Yes

Reviewer #2: Yes

4. Have the authors described where all data underlying the findings will be made available when the study is complete?

Reviewer #1: Yes

Reviewer #2: Yes

5. Is the manuscript presented in an intelligible fashion and written in standard English?

Reviewer #1: Yes

Reviewer #2: Yes

PLEASE NOTE: LINE NUMBERS IN AUTHOR RESPONSES REFER TO THE CLEAN MANUSCRIPT FILE

6. Review Comments to the Author

Reviewer #1: In this study protocol, the authors investigated the efficacy of NMES for mitigating the detrimental effect of hematopoietic stem cell transplant in patients with cancer. The application of NMES is patients with cancer is emerging and with a promising evidence. Therefore, new studies are welcome to have a comprehensive understanding of the efficacy and drawbacks of this strategy. I have included some comments for authors’ consideration as follows:

1. Line 71-74; The sentence needs correction.

Response to reviewer comment: 

We thank the reviewer for pointing out that these sentences require clarity. We have revised these lines accordingly, in combination with recommendation #2 below (now lines 66-69), which now state:

“For example, in a cohort of 201 HCT patients, only 2% were referred for physical therapy during the in-patient post-HCT recovery period (12). In addition, implementation of supervised exercise, like traditional resistance exercise, can be especially problematic for HCT patients who are immunocompromised and experience high levels of fatigue.”

2. Line 66-79 (Paragraph 2); In order to shorten the introduction, the information provided in the second paragraph could be summarized. Moreover, the structure and the information provided in the second paragraph of the introduction are as if exercise is the main intervention in this study while the authors just want to justify the use of NMES instead of exercise training. So, I recommend to re-structure the second paragraph and remove unnecessary information.

Response to reviewer comment: 

The authors appreciate the suggestion to summarize Introduction paragraph 2 to avoid distracting the reader. We have now revised this paragraph to by reducing the length while more concisely conveying our intended message, which was correctly interpreted by this reviewer. We have also made edits throughout the Introduction to be more concise while highlighting the main innovations of the current design, as suggested below in comment #3.

3. In the introduction, the authors have appropriately presented the current knowledge and understanding concerning the application of NMES in clinical populations such as cancer, however, it would be more interesting if the authors briefly highlight which specific gaps in the literature are covered in this study and what is the main novelty of this project.

Response to reviewer comment: 

We thank the reviewer for pointing out that the gaps in the literature are not obvious and should be detailed further. The following statement can now be found prior to the study goals (lines 100-104):

“While NMES may be safe and well-tolerated during HCT, its potential to benefit physical function and fatigue during HCT has yet to be tested in a randomized controlled setting. It is also unknown whether stimulating multiple lower body muscle groups and reducing prescription to three days/week to allow muscle recovery between sessions would induce a clinically meaningful impact on physical function during HCT.”

4. I congratulate the authors for providing the detailed information in the “Participants and study design” section.

Response to reviewer comment: The authors thank the reviewer for this generous comment.

5. Line 173; No informati

---

## [Decision Letter · Decision Letter 1]

17 Apr 2024

Neuromuscular electrical stimulation for physical function maintenance during hematopoietic stem cell transplantation: Study protocol

PONE-D-23-33264R1

Dear Dr. Anderson,

We’re pleased to inform you that your manuscript has been judged scientifically suitable for publication and will be formally accepted for publication once it meets all outstanding technical requirements.

Kind regards,

Masoud Rahmati

Academic Editor

PLOS ONE

Additional Editor Comments (optional):

Reviewers' comments:

Reviewer's Responses to Questions

**Comments to the Author**

1. Does the manuscript provide a valid rationale for the proposed study, with clearly identified and justified research questions?

Reviewer #1: Yes

Reviewer #2: Yes

2. Is the protocol technically sound and planned in a manner that will lead to a meaningful outcome and allow testing the stated hypotheses?

Reviewer #1: Yes

Reviewer #2: Yes

3. Is the methodology feasible and described in sufficient detail to allow the work to be replicable?

Reviewer #1: Yes

Reviewer #2: Yes

4. Have the authors described where all data underlying the findings will be made available when the study is complete?

Reviewer #1: Yes

Reviewer #2: Yes

5. Is the manuscript presented in an intelligible fashion and written in standard English?

Reviewer #1: Yes

Reviewer #2: Yes

6. Review Comments to the Author

You may also provide optional suggestions and comments to authors that they might find helpful in planning their study.

Reviewer #1: The authors have implemented all my comments appropriately. I have no further comments. The manuscript is acceptable for publication in the current format.

Reviewer #2: I am satisfied with the authors' responses and the resulting changes to the paper...................

7. PLOS authors have the option to publish the peer review history of their article (what does this mean?). If published, this will include your full peer review and any attached files.

Reviewer #1: **Yes: **Ehsan Amiri

Reviewer #2: No

---

## [Editor Report · Acceptance letter]

29 Apr 2024

PONE-D-23-33264R1 

PLOS ONE

Dear Dr. Anderson, 

I'm pleased to inform you that your manuscript has been deemed suitable for publication in PLOS ONE. Congratulations! Your manuscript is now being handed over to our production team.

Kind regards, 

on behalf of

Dr. Masoud Rahmati 

Academic Editor

PLOS ONE